# Development and Validation of a Targeted Metabolomic Tool for Metabotype Classification in Schoolchildren

**DOI:** 10.3390/metabo16010044

**Published:** 2026-01-04

**Authors:** Sheyla Karina Hernández-Ramírez, Diego Arturo Velázquez-Trejo, Eduardo Sandoval-Colín, Cristóbal Fresno, Mariana Flores-Torres, Ernestina Polo-Oteyza, María José Garcés-Hernández, Nayely Garibay-Nieto, Isabel Ibarra-González, Marcela Vela-Amieva, Guadalupe Estrada-Gutierrez, Felipe Vadillo-Ortega

**Affiliations:** 1Programa de Maestría y Doctorado en Ciencias Médicas, Odontológicas y de la Salud, Universidad Nacional Autónoma de México, Mexico City 04510, Mexico; sheyla-hernandez@facmed.unam.mx; 2Unidad de Vinculación de la Facultad de Medicina, Universidad Nacional Autónoma de México en el Instituto Nacional de Medicina Genómica, Mexico City 14610, Mexico; velazquez@ciencias.unam.mx (D.A.V.-T.); d.eduardo.sandoval@comunidad.unam.mx (E.S.-C.); mflores-torres@quimica.unam.mx (M.F.-T.); majogarces7@gmail.com (M.J.G.-H.); 3Programa de Maestría y Doctorado en Ciencias Biológicas, Universidad Nacional Autónoma de México, Mexico City 04510, Mexico; 4Red de Apoyo a la Investigación, Coordinación de la Investigación Científica, Universidad Nacional Autónoma de México, Mexico City 04510, Mexico; cristobalfresno@cic.unam.mx; 5Fundación Mexicana para la Salud, Mexico City 14610, Mexico; ernestinapoloo@gmail.com; 6Unidad de Bienestar Infantil, Hospital General de México Dr. Eduardo Liceaga, Mexico City 06720, Mexico; ngaribay@healthec.mx; 7Healthec By Tec Salud, Mexico City 10200, Mexico; 8Unidad de Genética de la Nutrición, Instituto de Investigaciones Biomédicas UNAM, Instituto Nacional de Pediatría, Mexico City 04530, Mexico; icig@biomedicas.unam.mx; 9Laboratorio de Errores Innatos del Metabolismo y Tamiz, Instituto Nacional de Pediatría, Mexico City 04530, Mexico; mvela@pediatria.gob.mx; 10Departamento de Inmunobioquímica, Instituto Nacional de Perinatología Isidro Espinosa de los Reyes, Mexico City 11000, Mexico; guadalupe.estrada@inper.gob.mx

**Keywords:** metabotypes, metabolomic profile, metabolomics

## Abstract

**Background:** Metabolomic profiling can uncover metabolic differences among seemingly healthy children, providing opportunities for personalized medicine and early detection of risk biomarkers for future metabolic disorders. This study aimed to identify and internally validate metabotypes in apparently healthy schoolchildren using targeted serum metabolomics and to assess the external validity of this metabotype classification tool in two separate groups of children. **Methods:** Data from schoolchildren aged 6–11 years were analyzed in two phases. In the first phase, we developed and validated a classification tool using targeted serum metabolomics in healthy children. Metabotypes were identified through unsupervised clustering with a self-organizing map, followed by assessment of cluster stability and classification accuracy. In the second phase, we tested the tool’s consistency by applying it to two additional groups: the same children from phase 1 after a 10-month physical activity intervention, and a separate group diagnosed with metabolic syndrome. **Results:** Three metabotypes were identified in healthy children: METBA (balanced profile), METLI (high lipid and glucose levels), and METAA (high amino acid levels). Internal validation showed strong cluster stability (ARI = 0.79) and high classification accuracy (0.95). After the intervention, 55% of children were reclassified, indicating diverse metabolic responses to physical activity. Among children with metabolic syndrome, 83% were classified as METLI and 13% as METAA. **Conclusions.** This tool revealed serum metabolomic diversity, enabling classification of healthy children into three distinct metabotypes. It also detects changes in metabotype classification associated with a physical activity intervention and identifies the majority of children diagnosed with metabolic syndrome within two groups. This supports the potential use of metabotypes as biomarkers and eventually for personalized interventions.

## 1. Introduction

The blood metabolome, reflecting metabolism, is a complex biological trait influenced by individual genomic makeup, the combined effects of epigenomic changes in relevant tissues, gut microbiota, circadian rhythms, and exposure to lifestyle-related environmental factors such as diet and fluctuating conditions that affect daily energy expenditure [1,2,3]. However, it remains the most accessible way to assess individual metabolic function, as circulating metabolites serve as indicators of tissue and organ capacity to process key metabolic substrates, such as glucose, fatty acids, and amino acids, through intermediary metabolic pathways [1,4]. Identifying variants of this biological trait, also known as metabotypes or metabolic phenotypes, which are groups of individuals with similar metabolic profiles, has been proposed to improve personalized nutrition and medicine [5,6,7]. By identifying distinct subgroups, metabotypes can help establish metabolic reference standards and identify metabolic risk profiles associated with various chronic non-communicable diseases [8]. Interestingly, using the blood metabolome as a biomarker of metabolic programming can help evaluate the early effects of environmental factors on individuals, especially during the first 1000 days of life. Exposure to harmful environmental influences during this period can cause lasting changes in the structure and function of organs and tissues, as proposed by Barker’s hypothesis [5,6]. Evidence suggests that factors like malnutrition and other components contributing to biological stress during this time induce physiological changes in the fetus or child, leading to phenotypes with a higher risk of developing chronic non-communicable cardiometabolic diseases in adulthood.

Identifying phenotypes in children can also be a valuable tool for early detection of risks for developing lifestyle-related diseases later in life and for evaluating targeted interventions focused on metabolic reprogramming.

From a clinical standpoint, identifying metabotypes in childhood may help detect metabolic traits early, before visible clinical signs appear, and support personalized lifestyle interventions.

However, the main obstacle to achieving this goal is the limited availability of pediatric metabolome data, which hinders understanding the functional implications of metabolic profiles beyond just the usefulness of specific metabolites for clinical risk assessment. Recent research has focused on identifying metabotypes associated with metabolic syndrome and childhood obesity [7,8,9,10]. This lack of data prevents a comprehensive understanding of how environmental and genetic factors influence the metabolome in early development. Characterizing the blood metabolome in children is a promising approach for evaluating these complex changes, which are influenced by environmental factors. Various techniques have been used to analyze metabotypes, primarily by analyzing venous blood [9,11].

This study’s primary aim was to identify metabotypes in a cohort of apparently healthy schoolchildren using targeted metabolomics and statistical modeling. The secondary goal was to validate the method using a second approach for metabotype classification of children and to apply this tool to differentiate and classify two distinct groups of children in which metabolomic changes were expected: children who participated in a physical exercise intervention and another group clinically diagnosed with metabolic syndrome.

## 2. Materials and Methods

### 2.1. Study Population

This study used prospectively gathered data from a group of schoolchildren aged 6 to 11 years who participated in a physical activity program during the 2012–2013 school year in public elementary schools in the State of Mexico, Mexico. Methodological details of this case/self-control study have been previously published [12]. This study was approved by the Research and Ethics Committee of the Faculty of Medicine at the National Autonomous University of Mexico (IRB 043-2012). It was authorized by local elementary education authorities (Educational Zone F029 in Ocoyoacac, Toluca, Estado de Mexico, Mexico). Written informed consent was obtained from parents or legal guardians.

### 2.2. Study Design

This tool development and validation study was divided into two phases. First, the metabotype classification tool was developed using targeted metabolomic data from blood samples of healthy schoolchildren, including internal validation. The second phase involved external testing of the metabotype classification tool by applying the pre-trained model to the same children from the initial phase, after a 10-month physical activity intervention, as well as to a separate group of children diagnosed with metabolic syndrome.

For the first phase of developing the metabotype classification tool, only healthy schoolchildren were included. Exclusion criteria for this group included a diagnosis of metabolic syndrome according to the requirements proposed by Cook et al. [13] or failure to meet the required fasting period of at least 8 h, as confirmed by the mother’s report and fasting glucose levels ≥ 100 mg/dL. Additionally, outlier metabolite values were used to exclude participants in all groups.

Samples from two new groups of children were included in the second phase, which aimed to assess the tool’s ability to differentiate metabotypes. The first group consisted of the same children from Phase 1 who had undergone a physical activity intervention, and the second group included schoolchildren from the same cohort diagnosed with metabolic syndrome. Participants who did not fast for at least eight hours were also excluded from both groups.

### 2.3. Sample Size and Effect

The study’s sample was determined by the availability of eligible participants with complete clinical and metabolomic data; therefore, a sample size calculation was not performed in advance. The statistical power of the available sample was calculated using a post hoc test, assuming a one-way ANOVA design with three metabotypes, a significance level of α = 0.05, and a total of 658 apparently healthy schoolchildren in Phase 1. Under these conditions, the study has over 90% power to detect small to moderate standardized differences in metabolite profiles between metabotypes (Cohen’s f ≈ 0.15, roughly equivalent to d ≈ 0.30 between any two groups). In the context of unsupervised clustering and validation, this sample size is therefore considered adequate for the methodological goals of developing and internally validating the metabotype classification tool.

### 2.4. Clinical Measurements

Weight, height, waist circumference (WC), and systolic/diastolic blood pressure were measured. Weight and height were obtained according to World Health Organization guidelines, and these measurements were used to calculate the BMI z-score for age [14,15]. The WC, as measured by Fernandez et al. [16], was used to calculate the waist-to-height ratio (WtHR). Blood pressure percentiles were classified by age and height according to the National High Blood Pressure Education Program guidelines [17]. All measurements were performed by trained and standardized personnel.

Peripheral venous blood samples were collected after an 8-h fast, and the serum was separated by centrifugation. Levels of glucose, triacylglycerols (TAG), total cholesterol (TC), high-density lipoprotein cholesterol (HDL-c), and low-density lipoprotein cholesterol (LDL-c) were measured in serum using an automated analyzer. Glucose, TAG, TC, and HDL-c were measured using standardized enzymatic methods, and LDL-c was calculated using the Friedewald method [18]. Atherogenic index (TC/HDL-c), TAG/HDL-c [19], LDL-c/HDL-c ratio [20], and TyG index [21] were also calculated.

### 2.5. Targeted Metabolomics

The targeted metabolome analysis included free carnitine (C0) and nine acylcarnitines (AC2, AC4, AC6, AC8, AC10, AC12, AC14, AC16, and AC18), 11 L-amino acids [alanine (Ala), glycine (Gly), arginine (Arg), methionine (Met), proline (Pro), valine (Val), leucine (Leu), phenylalanine (Phe), tyrosine (Tyr), citrulline (Cit), and ornithine (Orn)] and two amino acid metabolism-derived acylcarnitines (AC3 and AC5). Analysis was performed using a Quattro MicroAPI tandem mass spectrometer (MicroMass, Cary, NC, USA). All procedures for sample preparation and MS-MS analysis were performed using a NeoBase Non-derivatized kit (Perkin Elmer, Waltham, MA, USA) according to the manufacturer’s protocol. Serum samples were dried on filter paper, and a single 3-mm disk was punched per well. A mixture of stable isotope-labeled internal standards was added to each well. Results were expressed in μM. Samples were processed for metabolomics within three months of collection. The β-oxidation initiation rate [AC16 (μM)/C0 (μM)] and β-oxidation rate [AC16 (μM)/AC2 (μM)] were calculated.

### 2.6. Statistical Methods

Analyses were conducted using RStudio (version 4.2.2). Statistical significance was set at *p* < 0.05, and a Bonferroni correction was applied for multiple comparisons. Data distributions were assessed with the Anderson–Darling test and visual methods; homogeneity of variance was checked with Levene’s test. Variables were normalized when necessary, using the bestNormalize package.

Phase 1. Development and internal validation of the metabotype classification tool. Using baseline data from healthy schoolchildren, unsupervised clustering was performed with a Kohonen Self-Organizing Map (SOM). To reduce dimensionality and prevent overfitting, separate principal component analyses (PCA) were performed for amino acids and acylcarnitines, retaining components with eigenvalues greater than 1 (Kaiser criterion). Summary indices for each metabolite group were created as loading-weighted linear combinations of the original variables. The SOM was trained across grid sizes from 3 × 1 to 6 × 1 for 100 iterations, utilizing a rectangular toroidal topology and a Gaussian neighborhood. The optimal number of metabotypes was determined by the highest peak in the Silhouette coefficient. For internal validation (stability and separability): cluster stability and robustness were assessed by (a) calculating the Adjusted Rand Index (ARI) over 50 bootstrap resamples to quantify agreement of assignments under resampling; and (b) testing separability beyond chance with a supervised Random Forest (80/20 stratified split; features identical to the SOM input) trained to predict SOM labels, combined with a label-permutation null. SOM labels were permuted 250 times; for each permutation, the classifier was retrained, and its accuracy was measured. Observed accuracy is significantly higher than the permutation distribution suggests, indicating that the SOM partition identifies meaningful structure rather than random labeling. Outcomes that are deterministic functions of variables used to train the SOM (e.g., TyG from glucose and TAG; lipid ratios from lipids) were considered as descriptive characterizations across clusters rather than predictive endpoints for models trained on those same variables.

Phase 2. External evaluation of the metabotype classification tool. The metabotype classification of data from the same group of children in Phase 1, but after the 10-month physical exercise intervention, was analyzed using the pre-trained model.

The metabotype classification tool was also applied to data from a group of children diagnosed with metabolic syndrome. This group came from the same schools as the previous group but was excluded from the initial analyses due to their clinical diagnosis.

Additional analyses included comparing clinically significant physical activity-induced changes using paired *t*-tests or Wilcoxon signed-rank tests for continuous variables, and the McNemar–Bowker test for categorical variables. Comparisons among metabotypes were done with Pearson’s Chi-square or Fisher’s exact test for categorical data, and ANOVA, Welch’s ANOVA, or Kruskal–Wallis tests for continuous data, depending on the assumptions. Post hoc tests used were Tukey, Games–Howell, or Dunn tests with Bonferroni correction. Finally, baseline data from healthy schoolchildren served as a reference, and multivariate linear regression models were fitted to compare biomarker levels across metabotypes, adjusting for age, sex, BMI z-score, and WC.

### 2.7. Bias Control

Multiple strategies were used to minimize potential biases. Selection bias was mitigated by applying consistent eligibility criteria. Information bias was reduced through standardized clinical assessments performed by trained personnel and through uniform procedures for sample collection and processing. To reduce analytical bias and the risk of overfitting, dimensionality reduction was performed using PCA, and the stability and discriminative ability of the clustering solution were evaluated via internal validation using bootstrap-derived ARI and permutation-based Random Forest classification. Confounding during comparisons across metabotypes was controlled with multivariable models adjusted for age, sex, BMI z-score, and waist circumference.

## 3. Results

The sample from the cohort participating in the physical activity program, which included complete baseline and post-program metabolomic data, consisted of 1222 schoolchildren. Eighty-four were excluded due to incomplete data. Out of 1138 children assessed for eligibility, 203 (18%) were diagnosed with metabolic syndrome; these children were excluded from phase 1 of the study and later included in the exploratory validation phase. Additionally, 249 schoolchildren (22%) were excluded from phase 1 because they did not comply with the required fasting period. Twenty-eight children were excluded due to metabolite outlier results (Figure 1).

As a result, 658 healthy schoolchildren were enrolled in phase 1. Characteristics of this group are presented in Table 1.

### 3.1. Phase 1: Development and Internal Validation of the Metabotype Classification Tool

The initial selection of representative metabolites for modeling metabotypes was based on identifying the main contributors to variance. To reduce dimensionality and prevent overfitting, two indices were calculated. First, the two principal components of the amino acid PCA explained 85.5% of the total variance (PC1: 74.3%, PC2: 11.2%). PC1 was mainly influenced by Gly, Ala, Met, Val, Pro, Leu, Phe, and Tyr, while Arg primarily drove PC2. Using these nine loadings, the amino acid index (AAI) was created. For acylcarnitines, PC1 and PC2 (including AC2, AC4, AC8, AC10, and AC16) accounted for 74.37% of the variance (PC1: 53.05%, and PC2: 21.32%). The acylcarnitine index (ACI) was formulated using these five variables. Both indices were multiplied by −1 to ensure a positive correlation with metabolite levels.

The SOM model was trained on baseline data from children, including glucose, TAG, TC, C0, ACI (five acylcarnitines), AAI (nine amino acids), Cit, and Orn (urea cycle metabolites). These compounds also represent the primary groups of molecules involved in intermediate metabolism. A 3 × 1 neuron configuration was chosen as optimal based on the highest Silhouette coefficient of 0.35. Although this value is relatively low, it indicates the expected metabolic similarity among healthy children.

Internal validation demonstrated strong cluster stability, with an ARI of 0.79 across 50 bootstrap iterations. Additionally, a supervised Random Forest classifier trained to predict SOM-derived cluster membership achieved a classification accuracy of 0.95, significantly surpassing the mean accuracy of 0.42 from a null distribution created by permuting cluster labels. These findings support the robustness and non-random nature of the identified metabotypes.

The SOM model identified three metabotypes, based on the codebook vectors. Then, all variables were compared across the three metabotypes (Figure 2). The first group, or balanced metabotype (METBA), included 306 children (47%) and was characterized by symmetric levels of glucose, lipids, and amino acids. The second metabotype, or lipid-predominant metabotype (METLI), included 192 children (29%) who showed higher TAG, TC, LDL-c, and glucose levels. METLI exhibited the highest atherogenic index, TAG/HDL-c ratio, LDL-c/HDL-c ratio, and the TyG index (*p* < 0.001) (Figure 2D). Finally, the amino acid-predominant metabotype (METAA) included 160 children (24%) and displayed elevated AAI, higher Cit and Orn levels and a lower β-oxidation initiation rate. Other variables were similar across metabotypes.

Table 2 shows the characteristics of schoolchildren under basal conditions, stratified by metabotype. Age, weight, height, and blood pressure varied significantly among metabotypes. METBA consisted of the oldest children, while METAA had the lowest weight and height and the highest blood pressure percentiles.

Differences in all variables remained significant after adjusting for sex, age, BMI z-score, and WC. The adjusted model also showed notable differences in all acylcarnitine levels and the ACI.

### 3.2. Phase 2. External Evaluation of the Metabotype Classification Tool

The first external assessment group included children participating in a physical activity intervention. These children were the same as those studied in phase 1. As previously reported, overall adherence to the physical activity program was 91%. Overall, 55% of children changed their metabotype after the program (Figure 3). Among the children classified as METBA, 53% remained in the same group, 39% transitioned to METLI, and 8% to METAA. Transition from METBA to METLI was associated with increased TAG, TC, Orn, and C0 levels. The transition from METBA to METAA involved increases in AAI and urea cycle amino acids, as well as CO2, and a slight decrease in ACI.

Sixty-one percent of children remained in METLI after the intervention, 32% transitioned from METLI to METBA, and 7% transitioned to METAA. Moving to METBA was associated with decreases in TAG, TC, and glucose, whereas moving to METAA was associated with increases in Orn and AAI, along with decreases in TAG, TC, and ACI.

Ninety-one percent of children initially classified as METAA shifted to one of the other two groups; 47% moved to METLI, and 44% to METBA. Decreases in amino acids, C0, and TAG accompanied the shift to METBA. Meanwhile, those switching to METLI showed reduced levels of amino acids and C0, along with a slight increase in TAG (Figure 3).

The second group used to assess the tool’s effectiveness included 203 schoolchildren with metabolic syndrome. Of these, 162 (80%) were classified as METLI, 27 (13%) as METAA, and 14 (7%) as METBA.

Children classified as METLI had the highest TAG, TC, and LDL-C levels, as well as the highest atherogenic index, TAG/HDL ratio, LDL/HDL ratio, and TyG index (*p* < 0.001). In contrast, children with metabolic syndrome and classified as METAA showed markedly elevated amino acid levels and the highest AAI (*p* < 0.001). Meanwhile, METBA included the smallest proportion of children and displayed a more balanced biomarker profile

## 4. Discussion

Blood metabolome analysis provides a comprehensive yet insightful approach to understanding the complex, dynamic interactions among organs and tissues involved in metabolism. Environmental factors greatly influence this process. Our central hypothesis was that a single cross-sectional analysis of targeted blood metabolites could yield clear, meaningful information, enabling the characterization of groups with similar metabolic traits (metabotypes) among otherwise healthy children. In the future, this approach must be tested in its capacity to identify health outcomes associated with the initial classification prospectively.

Using unsupervised clustering of fasting serum levels of targeted metabolites, including glucose, TAG, TC, C0, ACI (five acylcarnitines), AAI (nine amino acids), Cit, and Orn (urea cycle metabolites), three metabotypes were identified in healthy Mexican schoolchildren, named METBA, METLI, and METAA. The key finding is that, despite all children being considered healthy, their circulating levels of glucose, lipids, and amino acids varied significantly, enabling the differentiation of three distinct metabotypes. These variations did not reach pathological levels and probably reflect differences in metabolite availability and utilization under normal physiological conditions.

Children in the METBA group exhibited lower levels of most metabolites, with no dominant signature. Although the children were older and taller, these features did not account for the metabolic differences between the groups, as indicated by regression analysis.

Children in the METLI group showed elevated levels of glucose, TAG, TC, and LDL-c. This profile indicates a preference for lipid energy sources. Notably, these children had the highest glucose levels, consistent with metabolic features seen in children living with obesity [8,22,23]. The METLI group did not display elevated circulating amino acids, unlike children with obesity. Despite no evidence of increased adiposity based on BMI, this metabolic state may predispose individuals to future dyslipidemia and insulin resistance, two hallmark features of metabolic syndrome and obesity [10,24].

Children classified as METAA exhibited the highest circulating amino acid levels, which may be linked to increased proteolysis, as observed during fasting to support gluconeogenesis; however, these children did not show elevated glucose levels [25]. Elevated urea cycle amino acids indicate active amino acid metabolism, while the lowest β-oxidation initiation rate suggests decreased fatty acid oxidation. Specific amino acid profiles, including higher levels of branched-chain amino acids (BCAAs) and aromatic amino acids (AAAs), such as those observed in METAA children, are established biomarkers of obesity and insulin resistance in both adults and children [7,8,26]. The selective reduction in L-arginine levels in METAA children coincides with increased blood pressure, potentially reflecting its use as a source of nitric oxide, a potent vasodilator [27].

Cardiovascular risk indices were calculated to assess the clinical significance of metabotypes. Although all metabolites and risk values fell within normal pediatric ranges, METLI exhibited the highest indices, indicating increased fatty acid and cholesterol metabolism and transport.

Internal validation of the classification model demonstrated high cluster stability and strong classification performance using a second classifier based on random forests, supporting the robustness of the identified metabotypes, which show direct biological coherence, making them suitable for evaluation as biomarkers for clinical use in the future.

In metabolomic clustering applied to pediatric populations, the moderate Silhouette coefficient found (0.35) should be considered in the context of normal children’s metabolomics, where metabolic profiles form a continuum, and some overlap among clusters is expected. In such datasets, Silhouette values between 0.25 and 0.45 are typical and do not suggest model inadequacy. Additionally, the targeted panel used, while suitable for the selected metabolites profiling, covers a significant but limited biochemical range, so we a priori accepted a reduced cluster separation. Collectively, these factors explain the moderate Silhouette value and support its biological plausibility.

**Model consistency.** We assessed the tool’s ability to classify children with two well-known conditions associated with metabolic changes [10,12,28]. First, the same children in phase 1 who completed a 10-month physical activity intervention were reclassified. Physical activity was associated with changes in metabolites across children’s metabotypes. Despite fluctuations in absolute metabolite concentrations after the intervention, the key variable balances for each metabotype remained stable, suggesting a robust metabolic structure. All metabotypes exhibited decreased lipid metabolism markers, although the patterns and magnitudes varied. METBA and METLI showed reductions in all circulating acylcarnitines, suggesting enhanced fatty acid β-oxidation, supported by a marked increase in the β-oxidation rate. In contrast, METAA showed reductions in some lipid metabolites, suggesting a limited lipid-oxidative response to physical activity, consistent with a lower AC16/AC2 ratio, an indirect indicator of β-oxidation.

At the same time, a moderate rise in several amino acids and the AAI was observed across all metabotypes, possibly indicating increased amino acid utilization for muscle remodeling. Meanwhile, alanine, the primary gluconeogenic amino acid, remained unchanged.

These descriptive findings challenge the traditional clinical view that healthy children have consistent metabolite levels within a normal range and highlight the potential for varied metabolic responses among children to a physical activity program.

According to reclassification, 45% of children retained their original metabotype, while 55% moved to different metabotypes after the physical activity intervention; however, not all metabotypes contributed equally to these changes. Notably, 91% of children initially classified as METAA changed metabotype, with about half shifting to each of the other two metabotypes, showing that this metabotype is more responsive to physical activity. METLI metabotype was the most stable. However, these changes should be interpreted cautiously, as the post-intervention use of the metabotype tool was exploratory and done without a randomized control group. METAA children had the highest BMI and BMI Z-scores after intervention; however, because body composition was not assessed, we cannot rule out the possibility that these changes are due to increased muscle mass or to natural growth or maturation.

Interindividual variability in response to exercise is well established, along with metabolic adaptations that include changes in glucose, lipid, and amino acid levels [28]. These responses differ considerably among individuals performing the same physical activity routines, emphasizing the need to understand these differences. Metabotypes may serve as potential modifiers of these responses, as the metabolic profile reflects the combined function of multiple tissues and organs. Variations in skeletal muscle, adipose tissue, and liver function likely contribute to the changes in metabotype classification observed after the intervention. These initial findings support the development of personalized interventions based on individual metabolic profiles.

A second group of children was used to evaluate the classifier’s ability to categorize children with metabolic syndrome. Notably, 80% of these children were classified as METLI and 13% as METAA; both patterns have been associated with children who have obesity and/or insulin resistance [29]. Based on these results, the classifier developed from healthy children can separate up to 93% of children affected by metabolic syndrome. These findings emphasize that, even within a seemingly homogeneous population, significant metabolic diversity exists, supporting the usefulness of metabotype-based strategies for risk assessment and personalized intervention planning.

**Clinical significance.** Identifying metabotypes in children is a crucial first step toward discovering biomarkers that could serve as epidemiological risk factors for disease. Evidence that using selected metabolites may provide a tool for this purpose was addressed in this paper. Future research should investigate how each metabotype relates to health outcomes to understand their functional importance. These biomarkers may help evaluate how early-life exposome modulation affects children’s metabolic health, providing insights into the mechanisms of metabolic programming during the first 1000 days and early childhood. The long-term impact of these biomarkers should be studied in prospective research. Such studies should utilize cohort-based designs, such as the CDMX-2000 days and OBESO cohorts [30], which gather detailed longitudinal social, clinical, and biomarker data from early pregnancy to age five.

Metabolomic profiling can be a valuable tool for early detection of children at risk of lifestyle-related metabolic diseases. Metabotype classification highlights the potential of circulating metabolites as markers of metabolic adaptations to exercise and encourages further research into hypotheses connecting early-life environmental exposures with metabolic plasticity.

This study has several limitations: 1. Children’s diet was neither controlled nor assessed, which limits understanding of its role in metabotype classification. 2. Body composition was not evaluated, despite fat mass being a known metabolic risk marker. It could be important for characterizing metabotypes, assessing how body fat distribution influences classification, and interpreting post-intervention changes. 3. Although the physical activity intervention in Phase 2 has been described elsewhere, its full details are not included in this manuscript, which limits the ability to evaluate exercise effects on the metabolome. 4. The lack of data on hormones, inflammatory biomarkers, and oxidative stress indicators restricts the scope of metabolic information and may reduce the clarity and biological interpretability of metabotypes. Additionally, a wider metabolome panel could improve the detection of metabolic effects. 5. External validation of the post-intervention sample was exploratory and conducted without a randomized control group, limiting the ability to determine intervention effects. 6. Key pre-analytical factors such as fasting duration, recent physical activity, hemolysis, batch effects, and specific quality-control procedures were not detailed. Since these factors can affect metabolomic measurements, the findings should be considered preliminary and require confirmation in larger, more diverse longitudinal cohorts with better pre-analytical controls.

## 5. Conclusions

This study identified three distinct metabotypes in children that, although not yet directly associated with clinical outcomes, could serve as early indicators of exposome effects and future metabolic risk. These metabotypes may also guide the development of personalized intervention strategies to support long-term metabolic health.

Future research should examine the potential association between these metabotypes and clinically significant metabolic outcomes to assess their predictive value over time. Additionally, implementing this tool with younger children and linking their metabotypes to environmental and behavioral exposures during pregnancy and the first 1000 days of life will help identify early-life factors that influence the development of specific metabolic profiles. This combined prospective and retrospective approach may reveal how early exposures impact metabolic phenotypes later in childhood.

From a clinical perspective, metabotype classification may support the early identification of children whose metabolic profiles reflect modifiable exposures. Because it is based on targeted metabolomics, this approach offers a feasible framework that could eventually guide personalized preventive strategies in pediatric populations.

## Figures and Tables

**Figure 1 metabolites-16-00044-f001:**
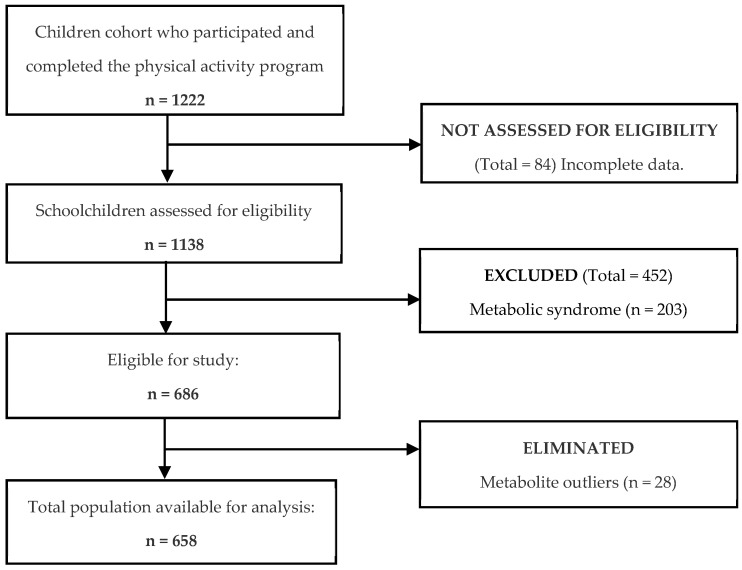
Flow chart of participants.

**Figure 2 metabolites-16-00044-f002:**
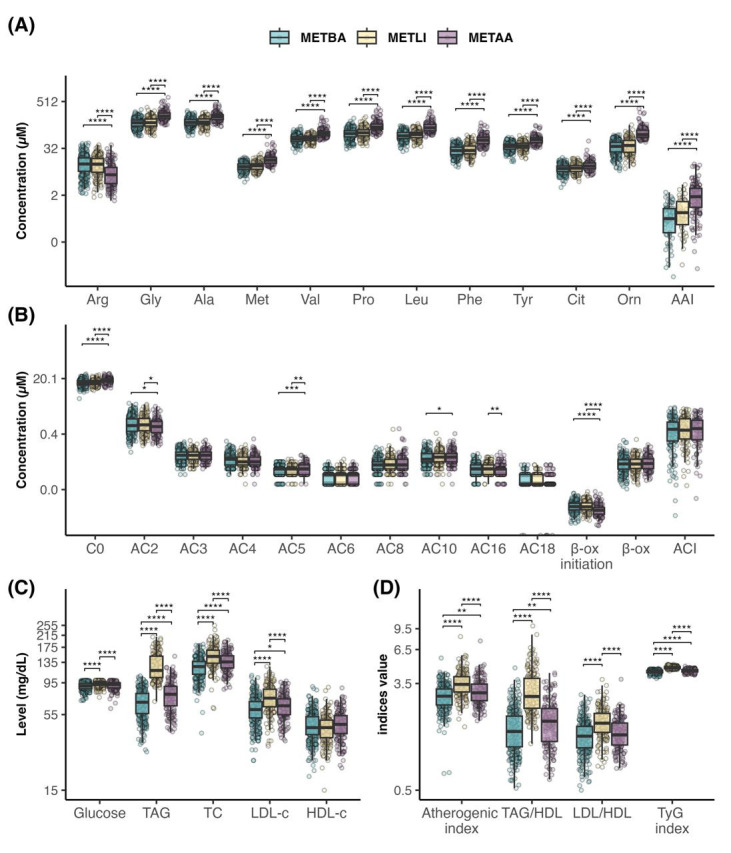
Baseline metabolite levels and cardiometabolic indices in schoolchildren by metabotype. Concentrations of (**A**) Amino acids and AAI, (**B**) Acylcarnitines, ACI, β-Oxidation initiation rate, β-Oxidation rate, (**C**) Other metabolites and (**D**) Cardiometabolic indices. Statistical significances is represented as * = *p* < 0.05, ** = *p* < 0.01, *** = *p* < 0.001, and **** = *p* < 0.0001.

**Figure 3 metabolites-16-00044-f003:**
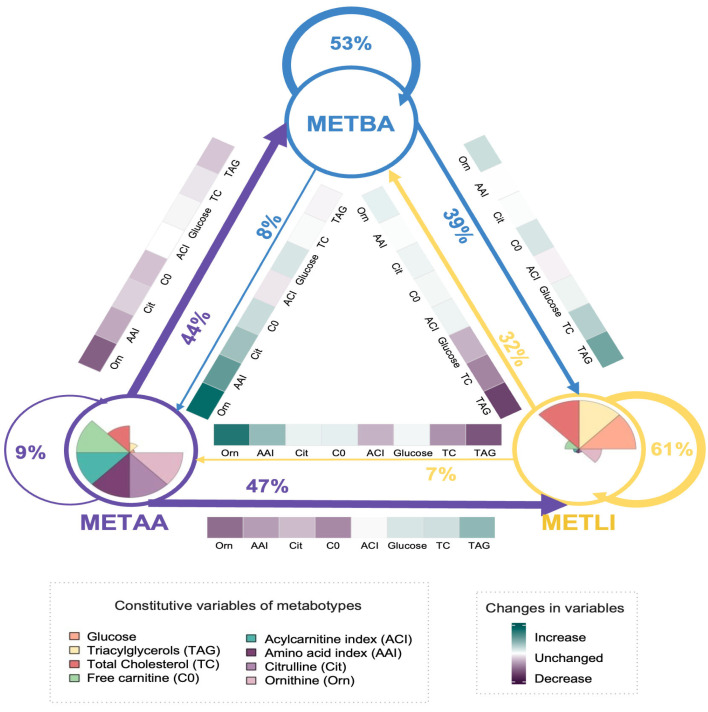
Classification transitions in children metabotypes after a physical activity program. Circles represent the metabotypes; colored segments show distinctive variables. Arrows indicate transitions after the intervention; their thickness reflects the proportion of children transitioning. Circular arrows show the proportion who remained in the same metabotype. Heatmaps display metabolite levels changes associated with each transition.

**Table 1 metabolites-16-00044-t001:** General characteristics of school children included in Phase 1.

Variable	Median (IQR) (*n* = 658)
Sex, *n* (%)	
Girls	337 (51)
Boys	321 (49)
Age (y)	8.97 (7.87, 9.90)
Weight (kg)	28 (23, 34)
Height (cm)	129 (123, 137)
Waist circumference (cm)	60 (55, 67)
Waist to height ratio (cm)	0.47 (0.44, 0.51)
BMI (kg/m^2^ )	16.40 (15.06, 18.80)
BMI for age (z score)	0.22 (−0.58, 1.15)
BMIfor age, n (%)	
Severe thinness	1 (0)
Thinness	10 (2)
Normal weight	454 (69)
Overweight	138 (21)
Obesity	55 (8)
Blood pressure (mm Hg)	
Systolic	105 (98, 112)
Diastolic	63 (56, 70)
Blood pressure (percentile)	
Systolic	75 (46, 90)
Diastolic	64 (46,84)

**Table 2 metabolites-16-00044-t002:** General characteristics of children in Phase 1 according to metabotype.

	METBA (*n* = 306)	METLI (*n* = 192)	METAA (*n* = 160)	*p*-Value ^a^
Sex, n (%)				
Girls	148 (48)	109 (57)	80 (50)	0.2
Boys	158 (52)	83 (43)	80 (50)	
Age (y) ^b^	9.27 (8.07, 10.02)	9.15 (8.03, 9.97)	8.21 (7.32, 9.37)	**<0.001**
Weight (kg) ^b^	28 (24, 35)	28 (24, 33)	26 (22, 31)	**0.004**
Height (cm) ^b^	131 (124, 138)	130 (123, 138)	126 (121, 133)	**<0.001**
Waist Circumference (cm) ^b^	61 (55, 69)	60 (55, 66)	59 (54, 65)	0.08
Waist to Height ratio ^c^	0.48 (0.05)	0.47 (0.04)	0.48 (0.05)	0.11
BMI (kg/m^2^) ^c^	17.28 (2.94)	16.81 (2.39)	16.78 (2.51)	0.2
BMI for age (z-score) ^c^	0.34 (1.28)	0.18 (1.11)	0.28 (1.16)	0.3
BMI for age, n (%)				
Severe thinness	1 (0)	0 (0)	0 (0)	0.05
Thinness	7 (2)	1 (1)	2 (1)	
Normal weight	191 (62)	147 (77)	116 (73)	
Overweight	79 (26)	31 (16)	28 (18)	
Obesity	28 (9)	13 (7)	14 (9)	
Blood pressure (mm Hg) ^b^				
Systolic	104 (97, 112)	104 (95, 111)	107 (101, 113)	**0.004**
Diastolic	62 (56, 70)	63 (56, 68)	64 (58, 71)	0.2
Blood pressure (percentile) ^b^				
Systolic	72 (46, 90)	71 (41, 88)	82 (65, 93)	**<0.001**
Diastolic	61 (43, 84)	63 (43, 82)	74 (50, 88)	**0.02**

^a^ Pearson’s Chi-squared test; Fisher’s Exact Test; Kruskal–Wallis rank sum test; One-way ANOVA; One-way analysis of means. Statistical significance was set at *p <* 0.05; ^b^ median (interquartile range); ^c^ mean (standard deviation).

## Data Availability

Deidentified clinical and metabolomics data collected for this study, along with a data dictionary defining each field in the dataset, will be made available to others beginning with the publication of this article. Additional related documents, including the study protocol, statistical analysis plan, and informed consent forms, will also be available. Similarly, the R scripts used for data preprocessing, clustering, and statistical analyses will be shared. Data and code will be provided upon request to the corresponding author by email, subject to the signing of a data and code access agreement and approval of a data and code use proposal.

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
