# Peer review of "Development and Validation of a Targeted Metabolomic Tool for Metabotype Classification in Schoolchildren"

_metabolites, 2026, doi:10.3390/metabo16010044_

Round 1
Reviewer 1 Report
Comments and Suggestions for Authors
The authors conducted a comprehensive study on the metabolomic profile of school-age kids, developed a model to help classify the metabotype, with great potential of early detection/intervention of metabolic diseases.
However, the authors need to address the following concerns before consideration for publication.
1. In the results and discussion sections, more detailed description on the three metabotype classification criteria should be given. What are the major differences among these three metabotypes? What are the implications for each metabotype? What does it mean when the metabotype changes after exercise?
2. Sex difference should be considered in the categorization of metabotype. Boys and girls might have different base line metabolome profile and can be studied separately.
3. The graphs presented are very confusing and hard to follow. I would recommend including only the significant statistics in the main paper to highlight key points and move the rest to supplemental information.
Author Response
Comment 1: In the results and discussion sections, more detailed description on the three metabotype classification criteria should be given. What are the major differences among these three metabotypes? What are the implications for each metabotype? What does it mean when the metabotype changes after exercise?
Response 1: Many thanks for your time and suggested modifications. An expanded description of the classification criteria was added in the Results and Discussion. After metabotype classification using the SOM modeling, all variables were compared between the three groups; significant characteristics of the three metabotypes are described in lines 363 to 382. At this point in our research, we are cautious about interpreting the clinical significance of these metabotypes. The presence of increased circulating lipids (METLI) may suggest an initial higher risk of metabolic complications in this group, but this must be evaluated in a longitudinal study. We have an ongoing cohort study that will provide the first insight into the clinical significance of these metabotypes. We included two groups of children in the second phase to test the tool's consistency in distinguishing children with other metabolic conditions, with no implications for the clinical meaning of these changes.
Comment 2: Sex difference should be considered in the categorization of metabotype. Boys and girls might have different base line metabolome profile and can be studied separately.
Response 2: No differences were observed between girls and boys across any of the variables used in this study.
Comments 3. The graphs presented are very confusing and hard to follow. I would recommend including only the significant statistics in the main paper to highlight key points and move the rest to supplemental information.
Response 3: We agree that both figures contain a lot of information, but we want to keep them since they may be helpful to readers. Table 3 was eliminated from the main manuscript and included as Supplementary Table 1.
Reviewer 2 Report
Comments and Suggestions for Authors
Dear authors,
Thank you for the opportunity to read and exchange scientific ideas. Please find my contributions attached.
Sincerely,

Author Response
Comments 1: It is strongly suggested that the study design(s) be incorporated into the title.
Comments 2: It is strongly recommended to align the objectives with those stated in the Introduction.
Comments 3: Avoid repeating the words and terms mentioned in the title.
Response 1, 2, and 3: Thank you very much for your thorough review. The title, abstract, and keywords were adjusted as suggested.
Comment 4: Lines 60-61: it is recommended to provide more detail on “process key metabolic substrates”.
Comment 5: Lines 71-73: it is recommended to clarify what type of stress is being referred to, metabolic or broader biological stress?
Response 4: The statements were elaborated.
Comment 5: It is suggested to explain the “clinical” justification for the study, where we could apply this knowledge from a biomedical perspective.
Response 5: An additional sentence was added between lines 98-100.
Comment 6: It is suggested to explain the main and secondary objectives of the study.
Response 6: This section has been expanded between lines 93 and 99.
Comment 7: It is recommended to describe the study design, based on the premise that several phases were carried out, as this would contribute to the methodological novelty. It is also recommended that the manuscript title should include the design in order to comply with international guidelines.
Response 7: The study design section was rewritten between lines 113 and 120.
Comment 8: Although the sample is a convenience sample derived from an existing cohort, with exclusions based on clinical and analytical criteria, it is strongly recommended that a sample calculation be included to determine the effect.
Response 8: A section on sample size was included in the Methods section.
Comment 9: The manuscript does not have an explicit section dedicated to bias control. It is recommended that the authors “formally” describe potential biases and the strategies adopted, in accordance with STROBE guidelines.
Response 9: A new section on bias control was included at the end of Methods.
Comment 10: It is recommended to add as a limitation the absence of a detailed description of the physical activity intervention in this manuscript. Although the authors mention that the protocol was previously published, the lack of information on the frequency, intensity, duration, and content of the sessions limits reproducibility and hinders the interpretation of the metabolic responses observed.
Response 10: Limitations of the study were expanded between lines 566 and 582.
Comment 11: We recommend adding two brief closing paragraphs, one on future prospects and study models that should be developed based on this one, and another paragraph addressing the clinical perspective and the feasibility of the method in practice.
Response 11: This suggestion was considered between lines 583 and 597.
Comment 12: It is strongly recommended to update with more than 50% of references between 2020-2025
Response 12: We conducted a thorough search of relevant references and included all of them in the initial version of the manuscript. Surprisingly, there are few recent references in this area of study.
Reviewer 3 Report
Comments and Suggestions for Authors
Dear Authors,
Your manuscript represents an ambitious and well-structured attempt to classify metabotips in a school population using targeted metabolomics and advanced cluster analysis methods. The work has significant potential and brings relevant insights into the metabolic profiling of children thanks to a clearly defined analytical panel, a standardized LC-MS approach, and a well-established statistical framework. However, certain segments of the manuscript — primarily those concerning the biological interpretation of metabotypes, control of key metabolic and preanalytical factors, and limitations of the study — require more detailed processing so that the paper is fully ready for publication.
First, without quantified information on the intake of proteins, fatty acids, carbohydrates and micronutrients, it is not possible to conclude whether metabotypes represent stable metabolic phenotypes or temporary changes caused by acute nutritional influence, knowing the fact that amino acids and acylcarnitines are extremely sensitive to nutrition. The above represents an essential biological factor that can influence the formation of clusters and not, as you state in the discussion, a marginal limitation. Therefore, it is necessary to clearly explain the impact of this deficiency in the part of the paper that refers to "research limitations"
Second, the lack of comprehensive measurements of body composition by bioimpedance represents not only a methodological limitation but also a biological one, because metabolomics and lipids are strongly related to adipose tissue in the part of quantification of visceral, adipose and total fat, which is crucial in pediatric metabolomics. Without these data, it cannot be clearly concluded whether the clusters represent differences in metabolism or only differences in fat distribution, which should be emphasized in the paper that the lack of these measurements is a biological limitation, not just a methodological one.
Thirdly, one of the key places in the manuscript that requires an amendment is related to the interpretation of the Silhouette coefficient. The value of Silhouette = 0.35 is stated, but no adequate explanation is given of what this value means in the context of biomedical data and why such a result is expected and acceptable. Silhouette values in the range 0.25–0.45 are not uncommon or problematic in biological and clinical datasets, especially in the pediatric population. Unlike engineering and synthetic data, children's metabolic profiles form a continuum rather than strictly separated categories, which naturally leads to partial overlapping of clusters. Therefore, the moderate Silhouette value actually reflects the biological reality — the heterogeneity of metabolism in the phase of growth and development — and not the weakness of the model.
Also, the metabolomic panel used was well chosen, but relatively narrow (limited number of amino acids, acylcarnitine and lipid indices). In metabolomics, broader panels encompassing hormones, inflammatory biomarkers, oxidative parameters, and extended lipid profiles typically yield stronger cluster separation. In this case, the limited width of the analytical spectrum naturally results in a moderate separation of metabotypes, which the authors should clearly and explicitly explain.
Fourth, As noted earlier, the panel is well selected but not broad enough for comprehensive metabolic characterization. A limited metabolomic panel narrows the breadth of metabotypes. Key biomarkers are missing - hormones (insulin, C-peptide, leptin, adiponectin), inflammatory markers (CRP, IL-6, TNF-α), oxidative stress and a wider set of fatty acids. Accordingly, the Authors should state that a wider panel can improve the separation and biological interpretation of metabotypes.
Fifth, external validation was conducted on a cohort of children included in a physical activity program without randomization and without a control group that was not exposed to the intervention. Therefore, it is not possible to exclude the influence of natural growth, maturation and development on the metabolomic profile and re-classification of metabotypes. The finding that 55% of children changed their metaboti¬p after the intervention should be interpreted very precisely in this context.
Finally, although technical procedures for metabolite analysis are standardized, data on preanalytical factors (time since last meal, physical activity before sampling, hemolysis, batch effects, quality controls) are not documented in detail. These factors have a huge impact on metabolomics. Given these limitations, the findings of this study should be interpreted as an initial validation of the concept of metabotypes in a school population, which requires further confirmation in larger, more diverse and longitudinal cohorts with an expanded biological panel and better control of preanalytical factors.
Conclusion: Major Revision - with adequate revision of the above points, the study has the potential to be highly valuable in clinical and population metabolic science.
Author Response
Comment 1: First, without quantified information on the intake of proteins, fatty acids, carbohydrates and micronutrients, it is not possible to conclude whether metabotypes represent stable metabolic phenotypes or temporary changes caused by acute nutritional influence, knowing the fact that amino acids and acylcarnitines are extremely sensitive to nutrition. The above represents an essential biological factor that can influence the formation of clusters and not, as you state in the discussion, a marginal limitation. Therefore, it is necessary to clearly explain the impact of this deficiency in the part of the paper that refers to "research limitations".
Response 1: We controlled for immediate nutritional effects on the serum metabolome by carefully selecting children who had fasted for at least 8 hours; most had fasted for 10 hours. However, we agree that the mid- and long-term effects of diet on the metabolome can be adequately evaluated only through dietary characterization. We addressed this limitation between lines 566 and 582.
Comment 2: Second, the lack of comprehensive measurements of body composition by bioimpedance represents not only a methodological limitation but also a biological one, because metabolomics and lipids are strongly related to adipose tissue in the part of quantification of visceral, adipose and total fat, which is crucial in pediatric metabolomics. Without these data, it cannot be clearly concluded whether the clusters represent differences in metabolism or only differences in fat distribution, which should be emphasized in the paper that the lack of these measurements is a biological limitation, not just a methodological one.
Response 2: The nature of the study from which blood samples were taken limited the ability to use relevant measurements such as body composition. We acknowledge this limitation between lines 566 and 582.
Comment 3: Thirdly, one of the key places in the manuscript that requires an amendment is related to the interpretation of the Silhouette coefficient. The value of Silhouette = 0.35 is stated, but no adequate explanation is given of what this value means in the context of biomedical data and why such a result is expected and acceptable. Silhouette values in the range 0.25–0.45 are not uncommon or problematic in biological and clinical datasets, especially in the pediatric population. Unlike engineering and synthetic data, children's metabolic profiles form a continuum rather than strictly separated categories, which naturally leads to partial overlapping of clusters. Therefore, the moderate Silhouette value actually reflects the biological reality — the heterogeneity of metabolism in the phase of growth and development — and not the weakness of the model.
Also, the metabolomic panel used was well chosen, but relatively narrow (limited number of amino acids, acylcarnitine and lipid indices). In metabolomics, broader panels encompassing hormones, inflammatory biomarkers, oxidative parameters, and extended lipid profiles typically yield stronger cluster separation. In this case, the limited width of the analytical spectrum naturally results in a moderate separation of metabotypes, which the authors should clearly and explicitly explain.
Response 3: This is an enlightening comment. Thank you. We reviewed and expanded the section on the study's limitations, described in lines 566-582.
Comment 4: Fourth, As noted earlier, the panel is well selected but not broad enough for comprehensive metabolic characterization. A limited metabolomic panel narrows the breadth of metabotypes. Key biomarkers are missing - hormones (insulin, C-peptide, leptin, adiponectin), inflammatory markers (CRP, IL-6, TNF-α), oxidative stress and a wider set of fatty acids. Accordingly, the Authors should state that a wider panel can improve the separation and biological interpretation of metabotypes.
Response 4: Our study methodology aimed to include compounds representing major metabolic pathways to maintain the integrity of the metabolic response while keeping them to a minimum; hence, we accepted a lower resolution of the response. We acknowledge this limitation.
Comment 5: Fifth, external validation was conducted on a cohort of children included in a physical activity program without randomization and without a control group that was not exposed to the intervention. Therefore, it is not possible to exclude the influence of natural growth, maturation and development on the metabolomic profile and re-classification of metabotypes. The finding that 55% of children changed their metabotipe after the intervention should be interpreted very precisely in this context.
Response 5: We proposed using both external groups of children to validate the tool's ability to classify individuals with altered metabolic conditions. No implications beyond the general description of these findings were intended with this initial approach. Results are promising, and we are now conducting a clinical trial with the proper design to evaluate these effects. We made several corrections to ensure proper interpretation of our results.
Comment 6: Finally, although technical procedures for metabolite analysis are standardized, data on preanalytical factors (time since last meal, physical activity before sampling, hemolysis, batch effects, quality controls) are not documented in detail. These factors have a huge impact on metabolomics. Given these limitations, the findings of this study should be interpreted as an initial validation of the concept of metabotypes in a school population, which requires further confirmation in larger, more diverse and longitudinal cohorts with an expanded biological panel and better control of preanalytical factors.
Response 6: We designed this study as part of a population-based research program on the effects of physical activity, involving about 20,000 children. Fieldwork for the program evaluation was conducted in four schools with 1,500 children, from whom we collected blood samples. We acknowledge that this approach limits control over many relevant variables and mention these limitations in the manuscript. We used as many criteria as possible to identify “healthy children” under fasting conditions, with sampling done early in the morning. More details about the study are available in a previous report (reference 16).
Round 2
Reviewer 1 Report
Comments and Suggestions for Authors
The revised version addressed the reviewers' concerns and greatly improved the readability of the manuscript.
It can be published as is.